



# Experimental study on the evolution of droplets size distribution during the fog life cycle

Marie Mazoyer[1], Fréderic Burnet[1], and Cyrielle Denjean[1]

[1]CNRM, Université de Toulouse, Météo-France, CNRS, Toulouse, France

*Correspondence to:* M. Mazoyer (marie.mazoyer@meteo.fr)

**Abstract.** The evolution of the droplet size distribution (DSD) during fog life cycle remains poorly understood and progress is required to reduce the uncertainty of fog forecasts. To gain insights into the physical processes driving the microphysical properties, intensive field campaigns were conducted during the winters of 2010–2013 at the Instrumented Site for Atmospheric Remote Sensing Research (SIRTA) in a semi-urban environment southwest of Paris city center to monitor the simultaneous variations in droplet microphysical properties and their potential interactions at the different evolutionary stages of the fog events. Liquid water content (LWC), fog droplet number concentration ($N_d$) and effective diameter ($D_{eff}$) show large variations among the 42 fog events observed during the campaign and for individual events. Our results indicate that the variability of these parameters results from the interaction between microphysical, dynamical and radiative processes. During the formation and development phases, activation of aerosols into fog droplets and condensational growth were the dominant processes. When vertical development of radiation fogs occurred under the influence of increasing wind speed and subsequent turbulent motion, additional condensational growth of fog droplets was observed. DSDs with one mode (around 11 $\mu$m) and two modes (around 11 and 22 $\mu$m) were observed during the field campaign. During the development phase of fogs with two droplet size modes, a mass transfer occurred from the smaller droplets into the larger ones through collision-coalescence or Ostwald ripening processes. During the mature phase, evaporation due to surface warming induced by infrared radiation emitted by fog was the dominant process. Additional droplet removal through sedimentation is observed during this phase for fog with two droplet size modes. Because of differences in the physical processes involved, the relationship between LWC and $N_d$ is largely driven by the droplet size distribution. Although a positive relationship is found in most of the events due to continuous activation of aerosol into fog droplets, LWC vary at constant $N_d$ in fog with large $D_{eff}$ (>17 $\mu$m) due to additional collision-coalescence and Ostwald ripening processes. This work illustrates the need to accurately estimate the supersaturation for simulating the continuous activation of aerosols into droplets during the fog life cycle and to include advanced parameterizations of relevant microphysical processes such as collision-coalescence and Ostwald ripening processes, among others, in numerical models.

## 1 Introduction

Fog is defined by the National Oceanic and Atmospheric Administration as a suspension of very small droplets in the air, reducing the visibility to less than 1 km close to the surface. These low visibilities are responsible for strong perturbation in aviation, transport, and health. The associated economic losses are estimated to several billions a year just for the airport (Gultepe et al.,



2017; Price et al., 2018; Kulkarni et al., 2019). Fogs are complex meteorological systems dealing with various fine scale processes. Continental fog often forms by radiative cooling of the surface (radiation fog) or by the lowering of pre-existing stratus to ground level (Tardif and Rasmussen, 2007). The fog life cycle is driven by radiation, turbulent, thermodynamic and cloud microphysics (hereafter referred to as microphysics) processes, which interact with each other in complex manners that are not

yet fully understood. In spite of significant advances in the skills of numerical weather forecast models (NWP) and Large-Eddy Simulation (LES) in recent decades, the timing of formation and dissipation of fog is poorly forecasted (Bergot et al., 2005; Van der Velde et al., 2010; Boutle et al., 2016; Martinet et al., 2020).

Accurate modeling of the fog requires precise determination of the microphysical parameters, such as liquid water content

(LWC), droplet number concentration ($N_d$) and effective diameters ($D_{eff}$). Both $N_d$ and $D_{eff}$ have a particularly large impact on the development of the fog layer due to their feedback on gravitational settling, LWC and radiative cooling at the fog top (Stolaki et al., 2015; Maalick et al., 2016; Boutle et al., 2018; Schwenkel and Maronga, 2018; Kutty et al., 2021). In situ measurements of fog microphysics have shown a large variability of these parameters: LWC, $N_d$ and $D_{eff}$ are commonly in the range 0.01-0.5 g.m$^{-3}$, 10-500 cm$^{-3}$ and 10-20 $\mu$m in diameter, respectively (Pilié et al., 1975; Choularton et al., 1981; Gerber,

1991; Wendisch et al., 1998; Liu et al., 2011; Lu et al., 2013; Niu et al., 2010; Price, 2011; Zhao et al., 2013; Gultepe et al., 2019; Liu et al., 2020). Large spatial and temporal variabilities have also been noticed during individual fog events, and even at different heights of fog layers (Okita, 1962; Pilié et al., 1975; Goodman, 1977; Pinnick et al., 1978; Garcıa-Garcıa et al., 2002).

The initial $N_d$ and $D_{eff}$ values depend on the ambient supersaturation and the aerosol population which acts as cloud

condensation nuclei (CCN) (Mazoyer et al., 2019). In recent years, various numerical studies investigated this aerosol indirect effect to study the influence of microphysics on fog life cycle (Boutle et al., 2018; Schwenkel and Maronga, 2018; Ducongé et al., 2020). Once the fog is formed, several physical processes affect the fog microphysical properties. Droplets can grow by water vapor condensation, by collision–coalescence (Xue et al. (2008) and Zhao et al. (2013)) or by Ostwald ripening that corresponds to the deactivation and evaporation of the smallest droplets in favour of vapor diffusion on the largest ones

(Wendisch et al., 1998; Boers et al., 2013; Yang et al., 2018). Droplets can conversely sediment by gravity (Bott (1991) and Degefie et al. (2014)) or turbulent motions (Tav et al., 2018), or evaporate if the supersaturation decreases due to heating or drying of the air mass, for example in the case of mixing with the residual dry air (Pilié et al., 1975; Choularton et al., 1981; Gerber, 1991). Schwenkel and Maronga (2018) showed in addition that different parametrizations of the activation and

condensation processes impact vertical extent and liquid water path of fog, which strongly affect the fog life cycle (Wærsted et al. (2019) and Karimi (2020)).

Observational studies showed that the fog cycle can be separated into four phases during which LWCs vary largely from phase to phase: the formation, development, maturity, and dissipation phases (Liu et al. (2011); Zhao et al. (2013); Lu et al. (2013)). Although these studies have contributed valuable insights on the physical processes driving the fog life cycle, mea-

surements of the evolution of fog microphysical parameters at different fog stages are currently lacking. As a result, current





NWP and LES models do not represent the microphysical processes explicitly and typically overestimate the observed LWC and $N_d$ (Mazoyer et al. (2017) and Boutle et al. (2018)). Recently, Boutle et al. (2021) pointed out model sensitivity of fog development to the shape of the cloud droplet size distribution. Improving our understanding concerning physical processes driving fog microphysical properties during the fog life cycle appears crucial for improving fog forecasting and mitigating the

impacts of such events.

Fog life cycle and microphysics are strongly related to dynamics (Mazoyer et al., 2017) and especially to the fog vertical development (Bergot, 2013). The radiation fog LES simulation of Boutle et al. (2018) shows that the gradual rise in the downwelling long-wave radiation, which causes the low transition towards a well mixed fog layer, is mainly driven by the fog

layer physical depth. Price (2019) pointed out the increasing wind speed as a non-local factor for fog development but they did not investigate its relation to microphysics. The present study explores the impact of the fog vertical development on its microphysical properties at the surface by taking advantage of the sampling of four fog events evolving from a thin layer to a thick fog developed vertically (Dupont et al. (2016) and Elias et al. (2018)).

In this study, we quantify the evolution of fog microphysical parameters of 42 fog events sampled at the Instrumented Site for Atmospheric Remote Sensing Research (SIRTA) downwind Paris urban area during the winters of 2010–2013. We aim to provide comprehensive information on the physical processes driving the fog microphysical parameters at the different fog stages, and how these processes affect the evolution of the fog life cycle. Specifically, in the indicated sections, the following questions are addressed :

1. Given the scarcity of data, what are the fog microphysics in the semi-urban environment of Paris ?

2. What are the dynamics conditions for fog formation and evolution ? What are the processes driving the vertical dispersion of fog and are the fog microphysics altered during the thin-to-thick transition?

3. What is the evolution of fog microphysics during the fog life cycle ? What are the key processes involved ?

4. What is the relationship between LWC and Nd ? How sensitive is this relationship to droplet size distribution ?

## 25    2    The data set

### 2.1    Observational site and instrumentation

The ParisFog field campaign was conducted at SIRTA (Haeffelin et al., 2005) located 20 km, south-west of Paris, France, in winter 2010 to 2013 in the framework of the ParisFog field campaigns (Haeffelin et al., 2010). During the winters of 2010 to 2013, specific instrumentation was deployed for the PreViBOSS project (Elias et al., 2012) to provide continuous observation

of aerosol and fog microphysics. Aerosol particle and droplet size distributions at ambient humidity were measured using a WELAS-2000 (Palas Gmbh, Karlsruhe, Germany, 0.4–40 $\mu$m) and a FM-100 (Droplet Measurement Technologies Inn., Boulder, CO, U.S.A., 2-50 $\mu$m). Both instruments are located on a scaffolding at about 2.5 m high, close to a PVM-100 from Gerber





Scientific inc. used as a reference for the LWC measurements in the size range of 2-50 $\mu$m.

Mazoyer et al. (2019) examined the properties of the hydrated aerosol particle and cloud droplet size distributions (DSD) during the fog formation of 23 events to evaluate the impact of aerosol particles on the fog microphysics. They derived accu-
rate estimations of the wet critical diameter for each case to exclude hydrated aerosols from the data analysis of $N_d$ and avoid subsequent overestimation of the activated droplet concentration. This method requires additional measurements such as CCN and dry aerosol size distribution. Here we extend the analysis to the 42 events with WELAS-2000 and FM-100 measurements and analyze the DSD characteristics during the entire life cycle. We used the median value of the wet critical diameters of 3.79 $\mu$m for all the cases. Sensibility tests have been performed and will be discussed in the discussion section. In the following,
data from the WELAS-2000 and the FM-100 are then combined on the [3.79-50] $\mu$m range diameter following the method described in Mazoyer et al. (2019).

The temporal and vertical evolution of the visibility were measured by two Degreanne diffusometers (DF20+ and DF320) located at 4 m and 18 m above ground. Confidence is given in FM-100 measurements by comparison of the integrated LWC
over its size range with the LWC measured by the PVM-100 and the visibility trend (Burnet et al. (2012) for the 2010-2011 period). Temperature and humidity sensors were located at heights between 1 and 30 m (1 m, 2 m, 5 m, 10 m, 20 m and 30 m) on an instrumented mast. Wind speed was measured by two ultrasonic anemometers at 10 and 30 m on the same mast.

## 2.2  Fog type and classification

During the three winter campaigns of 2010–2013, 42 fog events with reliable measurements of droplet size distribution and
meteorological conditions were retained in the analysis of the present study. Classification of the fog events was based on the measured visibility, temperature, wind speed, precipitation, cloud cover and ceiling height according to the Tardif and Rasmussen (2007) algorithm. At SIRTA, radiation fog and stratus-lowering fog occur with about the same frequency. Further classification of radiation fogs was based on their vertical development using the comparison of the two diffusometers: a thick fog produces low-visibility conditions at simultaneously 4 and 18 m, whereas a thin fog produces low-visibility conditions
at 4m only (Elias et al. (2009) and Dupont et al. (2016)). Development of radiation fog events from optically thin fog in a stable boundary layer to well-mixed optically thick fog were also observed during the field campaign. Among the 42 fog events analyzed here, 12 are radiative thick fogs, 6 radiative thin fogs, 4 thin-to-thick transition of radiation fogs and 20 stratus-lowering fogs.



# 3 Results

## 3.1 Overview of the fog microphysics

Statistics of $N_d$, LWC, and $D_{eff}$ values over life cycle of the 42 fog events are presented in Table A1 in the Appendix. The median values of $N_d$, LWC, and $D_{eff}$ vary over the ranges of 5-200 $cm^{-3}$, 0.002–0.096 g.$cm^{-3}$ and 8–22 $\mu$m, respectively,

which are in agreement with values reported for fog events in other regions (Eldridge, 1966; Pilié et al., 1975; Pinnick et al., 1978; Choularton et al., 1981; Kunkel, 1984; Gerber, 1991; Wendisch et al., 1998; Liu et al., 2011; Lu et al., 2013; Niu et al., 2010; Price, 2011; Zhao et al., 2013; Gultepe et al., 2019; Liu et al., 2020). Considerable variability on $N_d$, $D_{eff}$ and LWC is observed both for different fog events and during individual events. Additionally, fogs with a DSD with a single mode (around 11 $\mu$m) and two modes (around 11 and 22 $\mu$m) are observed. Determination of the number of the modes has been done for

each event according to the existence of local minima on the number DSD : 13 events have a bi-modal DSD that is about 30% of the 42 fog events analyzed here, and none of them are thin radiative fog. While droplet size distribution with two modes has already been observed by Frank et al. (1998); Wendisch et al. (1998); Gultepe and Milbrandt (2007), the origin of the largest second mode is still unclear.

Fig. 1 shows the relationship between $N_d$, $D_{eff}$ and LWC for the 42 selected fogs events. For a given LWC values range, $N_d$ decreases as $D_{eff}$ increases, except for very low LWC values < 0.01 g.$m^{-3}$. This trend is more pronounced as LWC range increases. Such a dependence between the size and the number of droplets is ubiquitous in convective clouds since droplets compete for the available water vapour. It appears less marked in fogs due to lower concentration values resulting from lower supersaturation. In addition Mazoyer et al. (2019) shown that mean Nd values averaged over the complete fog life cycle are

significantly lower than the Nd values values determined during the first hour of fog. Moreover, no direct correlation can be observed between $N_d$ and LWC. These results confirm that the predictability of droplet activation in fog can not be described by LWC only. Comparing the fog microphysics between the different classified fog events, lower $D_{eff}$ and LWC are observed in thin fogs (empty diamonds in Fig. 1) than in thick fogs (full symbols) for a given $N_d$. As a result the median value of LWC for thin fogs is only 0.010 g.$m^{-3}$ that is three times lower than for thick fogs, despite very similar median values of $N_d$ of

about 37 $cm^{-3}$. In addition, lower $N_d$ and LWC are observed for fogs with two droplet modes (round symbols) than for fogs with one droplet mode (diamond symbols). Indeed for the 2 modes group, the highest value of $N_d$ and LWC reach 53 $cm^{-3}$ and 46 g.$m^{-3}$ respectively, for F2 in Table A1. In contrast among the 1 mode group, there are 11 cases with higher $N_d$ and 8 cases with higher LWC, and they are not necessarily the same cases. This puts forward the need of a fog microphysical analysis during its life cycle.


In the following, our analysis of the evolution of the fog life cycle takes advantage of the occurrence of the four episodes of thin-to-thick radiative fog transition to explore the impact of the fog vertical development on the microphysics. We present results for a typical event representative of the general behaviour of the other thin-to-thick events. In a second part, two thick events are analyzed in details to provide guidance on a statistical analysis of the whole set of events.



## 3.2 Thin-to-thick transition

Fig. 2 presents the visibility and thermodynamics (temperature, wind speed and direction) evolution for the F2 thin-to-thick fog event. Temperature at 10 m decreases all along the fog event. Wind direction is constant from south east during the event and wind speed is under 2 m/s till 2250 UTC. At 2130 UTC the visibility at 4 m decreases under 1000 m. However because of the weak wind speed (< 2m/s), fog is not able to develop on the vertical (Rodhe, 1962; Duynkerke, 1991). At 2250 UTC wind speed increases to around 3 m/s with maximum of 5 m/s. At 2300 UTC this increase of the wind speed is immediately followed by the vertical development of the fog marked by a decrease of the visibility at the 18 m altitude level. Finally, fog dissipates at 0540 UTC simultaneously at 4 and 18 m. The three others thin-to-thick transitions sampled during this campaign were also concomitant with an increase in wind speed. Therefore, in agreement with Bergot (2013); Price (2019); Gultepe et al. (2021) studies of fog dynamics, the thin-to-thick transition is most likely caused by the increase of wind speed and the subsequent turbulent motion.

The temporal evolution of the fog microphysics properties of F2 are reported on Fig. 3. Before the vertical development (red lines), the fog layer appears discontinuous with alternating periods of dense fog and almost clear air despite the visibility close to the ground (4 m) remains below 1km. FM-100 measurements indicate that both $N_d$ and LWC fluctuate strongly with values ranging between 5-220 cm$^{-3}$ and 0.01-0.11 g.m$^{-3}$ respectively, while $D_{eff}$ remains more stable $\sim$ 12 $\mu$m. Such a feature of stable $D_{eff}$ could reflect the inhomogeneous mixing with residual air (Baker et al., 1980). Importance of mixing in fog is supported by observational studies of fog top by Pilié et al. (1975); Choularton et al. (1981); Gerber (1991). However wind and turbulence are very low during this thin phase and then it is rather expected homogeneous mixing in such stable layer. It is then more likely that such fog patches, that are often observed in stable conditions, actually reflects that condensation of liquid water firstly occurs in isolated layers where the temperature reaches locally the dew point. In contrast the fog layer becomes more continuous when it thickens vertically. There are still large fluctuations on 1s samples (black lines), especially on LWC and $N_d$, but minute average data (green lines) reveal gradual changes that are relatively slower. After a short increase of LWC during 20 min, it seems to fluctuate slowly around a steady state, while $N_d$ decreases almost continuously and therefore $D_{eff}$ follows a reverse trend.

The corresponding composite particle size distribution derived from the combination of WELAS-2000 and FM-100 measurements are displayed on Fig. 4. During the thin phase (red lines and orange shaded area) no mode appears on the size distribution that exhibits a continuous decrease for supermicronic particles, as reflected by the median and 25th-75th percentiles. There are only a very small fraction of samples containing large droplets as revealed by the 95th percentile. On the opposite during the thick phase (green lines and shaded area) we observe a first droplet mode centered at 5 $\mu$m, and large amount of large droplets with a second mode centered around 20 $\mu$m with very low dispersion around the median values. This could be explained by the growth of the cloud droplets by condensation and collision-coalescence processes. Indeed Mazoyer et al. (2019) found for this event a wet critical diameter of 4.24 $\mu$m. It follows that during the thin phase, particles are mainly



deliquescent aerosols with very few activated particles. On the opposite once the fog is vertically developed, numerous droplets have been produced by activation and subsequent growth by condensation. They can then reach the threshold diameter required to trigger the collision-coalescence process. Shortly after the vertical development we observe the shift of the droplet size distribution toward largest sizes with droplet as large as $\sim 30~\mu$m. The thin-to-thick transition is followed on Fig. 3 by a slight

decrease of $N_d$ which confirms the possible occurrence of collision-coalescence. The enhanced turbulence during the transition may favour the onset of collision-coalescence for small droplets diameter according to Xue et al. (2008). The thickening of the fog layer will obviously also increase the opportunity for gravitational settling droplets to collect more water along their path as they sediment. The largest mode of fog droplets is not observed before the fog vertical development, most likely due to the very low rate of aerosols activation in fog patches which prevents the growth of fog droplets to sufficient size.

The three other fog events with thin-to-thick transition exhibit very similar microphysical properties and temporal evolution. They reveal that thin fogs are composed of mainly unactivated particles while when the thickness increases large cloud droplets appear in very short time scale that suggests growth by condensation and collision-coalescence. In the following we investigate the processes occurring at 4 m in fog already developed on the vertical.

**3.3 Meteorological conditions during thick fog events**

Fig. 5 shows the evolution of the visibility, temperature and wind speed during two contrasting thick fog events. F9 is a 12h long radiative fog formed at 150 m before reaching the ground and characterized by a single droplet mode. As underlined by Stolaki et al. (2015) and Mazoyer et al. (2017), the formation of fog low in altitude is very common at SIRTA and 88 % of the radiation fog events during the field experiment followed a similar pattern. F32 is a 6h long radiative fog formed at ground

characterized by two droplet modes. As shown in Fig. 1 (see the red circles), these two fogs are representative of the ensemble of events in terms of median microphysics values during the whole life cycle.

For both cases, temperatures in the first 30 m above the ground reached maximum values around 14 UTC the day before, then radiative cooling occurred in the afternoon and during the night, with a continuous decrease of the temperatures, stronger

near to the surface. For both fogs, the visibility show sudden and simultaneous fog formation at 4 m and 18 m. At the same time, temperatures become even and decrease much slowly until they reach a minimum in the morning. Although less apparent in the figure due to the log scale, the visibility follow the same trend until a minimum reached at around the same time. Such a minimum had also been put forward by Pilié et al. (1975), according to them it is associated to a slow droplet evaporation. A minimum on temperature evolution during fog life cycle had also been observed by Price (2011) but they did not link it to

visibility evolution. All the 42 fog events sampled during the experiment exhibit similar temporal evolution of visibility and temperature at 2 m, as well as of the upwelling long-wave radiation flux at 10 m. Fig. 6 shows that the minimum visibility occurs almost simultaneously with both the minimum temperature and the minimum upwelling infrared flux. This appears to be consistent with a strong influence of surface warming on the fog life cycle due to infrared radiation emitted by fog, which would causes in turn a slow evaporation of fog droplets. The impact of the short wave warming can be rule out since for 32



events the mimimum of temperature occurs before the sun rise. Note that some scatter exists, with delays of up to several hours, when advection processes are involved. After a slight increase, the visibility steadily increases and fog dissipates at the surface at 09 UTC and 13 UTC for F9 and F32, respectively.

### 3.4 Temporal evolution of microphysical properties

In order to investigate the fog temporal evolution, a division in four phases depending on the visibility evolution is commonly made (Pilié et al., 1975; Niu et al., 2010; Liu et al., 2011; Zhao et al., 2013; Degefie et al., 2014). Pilié et al. (1975) had shown that droplet concentration and LWC reach maximum values during the fog life cycle visibility minimum and Liu et al. (2011) linked their visibility division to turbulence evolution which is also a commonly used parameter to divide fog into phases (Nakanishi, 2000; Porson et al., 2011; Bergot, 2013). In the following, each fog event is separated into four phases based on

the evolution of visibility calculated with a 15 minutes sliding average (see color-time splitting in Fig. 7). The formation phase (red line) is characterized by a sharp decrease of the visibility from 1000 m to 200 m in about 30 min for both the cases. During the development phase (green), the visibility continues to decrease but very slowly until its minimum value : for F9 the visibility lost only 50 m in 4h15. During the mature phase (yellow), the visibility slightly increases with a similar rate. Finally during the dissipation phase (blue), the visibility increases rapidly to 1000 m. Consistently, comparable trends are depicted for

microphysical properties, especially for droplet concentration and LWC, with sharp variations during formation and dissipation phases, and weaker increase and decrease during development and mature phases, respectively. It is obvious that there are large fluctuations within each phase, because many processes impact locally the microphysics, but we consider here the general trend in a attempt to characterise the typical life cycle.

A statistical characterization of the microphysical properties during the 4 phases is performed on the 42 events. Fig. 8 and Table 1 show the linear regression slopes of the temporal evolution of LWC, $N_d$, $D_{eff}$, and temperature at 2 m during each phases. The formation phase is characterized by a strong increase of LWC, $N_d$ and $D_{eff}$, associated to a decrease of the temperature. The cooling of the air masses resulted in the condensation of water vapor and the activation of aerosols into fog droplets. New fog droplets are still formed in the development phase, causing a slight increase of $N_d$ and $D_{eff}$ although their

production and growth are much slower. Such increase of $N_d$ during this phase is surprising as condensation is expected to consume ambient supersaturation and consequently limit new aerosol activation. During the mature phase, the temperature becomes positive and both $N_d$ and LWC decline. At the dissipation phase, the decrease of $N_d$ and LWC is more pronounced.

Fogs issued from a stratus lowering (blue squares in Fig. 8) experience less cooling for activation than radiative fog, as pre-

viously reported by Dupont et al. (2016). As a result, $D_{eff}$ and LWC evolve slower for fogs issued from a stratus lowering than for radiative fogs during the formation and development phases. Dupont et al. (2016) hypothesized that formation of fog from stratus lowering is due to the droplets sedimentation and evaporation, which induce the cooling below the stratus base and the enhancement of activation/condensation processes. Koračin et al. (2001) suggested that radiative cooling at the stratus top and large scale subsidence are responsible for the mixing of the dryer layer under the stratus base, which favours its descent. These



hypothesis contrast with the formation mechanism of radiation fogs (ground surface cooling) and could explain the differences in microphysics of radiation fog and fogs issued from stratus lowering.

Variabilities between fogs with a single and two droplet modes are stronger during the formation phase. The gradient of $D_{eff}$ during the formation phase is slower and even sometimes negative for fogs characterized by two droplet modes compared to those with a single droplet mode. To gain insights into these differences, Fig. 9 shows the number size distribution of aerosols particles and fog droplets during the two contrasted fog events presented in section 3.3. Observations indicate that the size distribution of fog droplets vary differently over time depending on the initial fog microphysical properties. For our 42 cases, the number of droplet modes is determined during the formation phase. When fog is characterized by a single droplet mode (F9), the number size distribution can be approximated by a bimodal log-normal size distribution with an aerosol mode centered at 0.4 $\mu$m and fog droplet mode centered at 11 $\mu$m. Droplet size distribution is marked by very little increase in $D_{eff}$ during the development phase. During the mature phase, $D_{eff}$ declines and the number concentration of hydrated aerosols grows. This can be due to the release of water vapor from the start of droplet evaporation due to surface warming (ie. section 3.3 and Fig. 8). At the dissipation phase, $N_d$ declines but the modes of hydrated aerosols and droplets subside maybe under the influence of sedimentation processes. When compared to F9, F32 shows an additional droplet mode centered at 22 $\mu$m formed at the same moment as the droplet mode centered at 11 $\mu$m. The observed decrease of $D_{eff}$ in Fig. 8 during the formation phase is due to the faster increasing concentration of the smaller droplet size. During the development phase, droplets size distributions exhibit a drastic rise in number of the largest droplet and a decrease of the smallest one. This mass transfer phase can be explained by either collision-coalescence process or Ostwald ripening process as put forward in Wendisch et al. (1998); Boers et al. (2013); Yang et al. (2018). The decrease of $N_d$ and $D_{eff}$ begins during the mature phase, but more quickly than in F9. Since the sedimentation rate of droplets increases with droplet diameter, it is likely that the droplet removal through sedimentation is accelerated in F32 compared to F9. Fig. 7b shows the evolution of the $95th$ percentile of droplets diameter (top dashed lines). Quasi periodic oscillations of this diameter are observed during the mature phase. Bott (1991) had also observed such fluctuations using numerical modelling and suggested that these fluctuations could be due to the combined effect of condensation and sedimentation processes of the largest droplet mode. During the dissipation phase, the droplet mode centered at 22 $\mu$m vanishes probably due to the combination of evaporation and sedimentation processes, leading to fog droplets distributed in a unique mode before the total dissipation of the fog event (Fig. 9).

Our statistical analyzes highlight the main presumed microphysical processes during developed fog: the activation/condensation and evaporation/de-activation, the sedimentation, the collision-coalescence and the Ostwald ripening process Wendisch et al. (1998); Boers et al. (2013); Yang et al. (2018).

### 3.5 Correlation of $N_d$ and LWC

In an attempt to go further in the microphysical processes examination, we investigate the link between the rate of LWC time increase with $N_d$ (slope value of $LWC = aN_d$) and the correlation coefficient that relates this evolution. A low slope value





suggests strong droplets growth by condensation/evaporation compared to new droplets activation/de-activation. A low correlation coefficient value suggests that this evolution is far from being linear so that other processes than activation/condensation and evaporation/de-activation occur. Fig. 10 shows the slope value of $LWC = aN_d$ time evolution against the correlation coefficient associated to this evolution for the 42 fog events at the different phases. In general, $N_d$ positively correlates with

LWC with a strong correlation factor of around 0.8. For 8 events, the correlation factor between $N_d$ and LWC is poor with values lower than 0.7. Half of these cases are characterized by $D_{eff}$ larger than 17 $\mu$m (green points in Fig. 10). The other half cases are characterized by two droplet modes (diamonds markers). Fig. 11 shows the 5 minutes averaged $N_d$ as a function of LWC for our two contrasting fog events characterized by a single (F9) and a double droplet mode (F32), respectively. On Fig. 10 (see the grey circles), $N_d$ positively correlates with LWC for F9, whereas the correlation is poor for F32. In section

3.4, we showed that collision-coalescence process or Ostwald ripening process may had occur during F32, while it is more unlikely during F9. Also we showed that F32 may had more droplets removal through sedimentation than F9. Altogether, our observations indicate that the linear relationship between LWC and $N_d$ is highly dependent on $D_{eff}$. A positive correlation is found when fog droplets exhibit lower sizes. A likely explanation is that fogs with higher $D_{eff}$ experience more sedimentation and collision/coalescence processes.

These results clearly contradict with observations in adiabatic cloud core where LWC vary with constant $N_d$ (Rosenfeld and Lensky (1998) and Pawlowska et al. (2006)). The increase of LWC with increasing $N_d$ in most of the fog events is mainly due to the continuous activation of aerosol into droplets (i.e. section 3.4). When compared with a cloud, fog is usually formed under conditions with lower supersaturation values (Mazoyer et al., 2019). It is plausible that the low supersaturation limits the

growth of droplets by condensation and the consumption of the water content. The excess water vapor could therefore become available for additional activation of aerosols into cloud droplets.

## 4   Discussion

The results obtained in this study were obtained by analyzing DSD calculated with the unique combination of a WELAS-2000 and a FM-100. We followed the method presented in Mazoyer et al. (2019) to take advantage of both instrument limitations

and resolution. A unique wet critical diameter of 3.79 $\mu$m has been then used for all our events. That means that we only consider particles larger than 3.79 $\mu$m. Previous observational study of fog evolution commonly used the [2-50] $\mu$m diameter range (Wendisch et al., 1998; García-García et al., 2002; Gultepe and Milbrandt, 2007; Niu et al., 2010; Price, 2011; Liu et al., 2011; Lu et al., 2013; Zhao et al., 2013). However they were not able to assess the wet critical diameter and may have considered hydrated non-activated particles that would result in an increased concentration. To improve the representation of

fog microphysical processes in numerical weather prediction models both species must be studied separately. Recently Boutle et al. (2018) have shown that this is of crucial importance and found a size limit of 6 $\mu$m for their specific LES case study. The wet critical diameter used was the median value determined over 23 fog events of the 42 studied here by Mazoyer et al. (2019). The variation range varies from 3.03 and 4.67$\mu$m for the $25^{th}$ and $75^{th}$ percentiles. Thus we may have neglected droplets





or considered hydrated non-activated aerosols for some of our fog event. To validate this approximation the linear regression slopes of LWC, $N_d$, $D_{eff}$ calculated using this mean wet critical diameter has been compared to the values calculated using the wet critical diameter determined individually for 23 fog events by Mazoyer et al. (2019). The linear regression slopes agree well with the two approaches (see Fig. A1 in appendix), which suggests that, for what concerns our processes analyzes, the

concentration and diameter of fog droplets can be well estimated using a single wet critical diameter.

## 5  Conclusion

This paper presents in-situ observations of meteorological and microphysical properties for 42 fog events that occurred down-wind Paris urban area during the winters of 2010-2013. The analysis separates the fog events between their mechanism of

formation, their vertical development and their evolutionary stages in order to examine the physical processes driving the fog life cycles. The median values of $N_d$, LWC, and $D_{eff}$ vary over the ranges of 5-200 cm$^{-3}$, 0.002–0.096 g.cm$^{-3}$ and 8–22 $\mu$m, respectively, which is in agreement with values reported for fog events in other regions. Variabilities in these parameters between the events and for individual events are attributed to the combination and interaction of microphysical dynamical, radiative processes and surface conditions.

During the fog formation phase, activation of aerosols into fog droplets and condensational growth are the dominant processes. The former process is responsible for the formation of smaller droplets, whereas the latter one is responsible for the growth of the larger droplets by condensation of water vapor due to the cooling of the air masses.

Approximately 10 % of the events remain as optically thin fog, whereas 90 % form optically thick fog. The increase of the

wind speed and the subsequent turbulent motion has an important role in the vertical development of the fog. When compared to thick fogs, thin fogs display lower $D_{eff}$ and $N_d$ due to the presence of a residual dryer layer that counteracts their growth. In the thin-to-thick transition, additional vertical mixing of air masses causes the growth of fog droplets by collision-coalescence or condensation.

The initial droplet size distribution has a strong influence on the evolution of fog microphysical properties over time. Fogs with a single ($D_{eff}$ around 11 $\mu$m) and a double ($D_{eff}$ around 11 and 22 $\mu$m) droplet mode are observed during the formation phase. At the development phase, most of the observed fog events experience a slight continued production and growth of fog droplets by activation and condensation processes. When fog is characterized by two modes, a mass transfer occurs from the smaller fog droplets into the larger droplets likely due to collision-coalescence or Ostwald ripening processes. During the

mature phase, evaporation due to surface warming induced by infrared radiation emitted by fog is the dominant processes. Additional droplet removal through sedimentation is observed during this phase when fog events are characterized by two modes. Because of differences in the physical processes involved, the relationship between LWC and $N_d$ is largely driven by the droplet size distribution. Although a positive relation is found in most of the events due to continuous activation of aerosol

into fog droplets, LWC vary at constant $N_d$ in fog with large $D_{eff}$ (>17 $\mu$m) due to addition collision-coalescence and Ostwald ripening processes.

Our results show that the droplet size distribution has a large impact on the development of the fog layers due to its feedback on the physical processes driving fog life cycle. The current NPW and LES models rely on bulk formulations of integral values (e.g. LWC and $N_d$) or assume a droplet size distribution fixed in space and time to represent fog microphysical properties (Hong and Lim, 2006; Seity et al., 2011; Khain et al., 2015; Vié et al., 2016) but recent intercomparison of radiation fog models (Boutle et al., 2021) put forward the fog sensitivy to the shape of the cloud droplet size distribution. Explicitly simulating the changes of the droplet size distribution in the fog layer by including an advanced parameterizations of all relevant microphysical processes such as size-resolved collision-coalescence and Ostwald ripening, among others, could significantly improve fog forecast models that often suffer from too high values of LWC (Philip et al., 2016; Pithani et al., 2019; Ducongé et al., 2020). This study also showed that fogs experience continuous activation during formation and development phase. It is not consistent with the saturation adjustment often used in numerical models and use of different supersaturation parameterisation has been shown to impact the fog life cycle (Schwenkel and Maronga, 2018; Gultepe et al., 2021; Boutle et al., 2021). This highlight the importance of a careful computation of fog supersaturation during fog evolution.

**Data availability**

Advanced data are available in table A1. Data are available upon request to the authors.

**Author contributions**

This work is a part of the PhD of MM supervised by FB. MM, FB and CD analyzed the data and wrote the paper.

**Competing interests**

The authors declare that they have no conflict of interest.

**Acknowledgements**

The authors are very grateful to all SIRTA operators and database managers. This campaign was held in the framework of the PreViBOSS project, supported by DGA/DGIS. This research was partially funded by the European Commission's Seventh Framework Programme (FP7/2007-2013) under the SESAR WP 11.2.2 project, under grant agreement 11-120809-C.





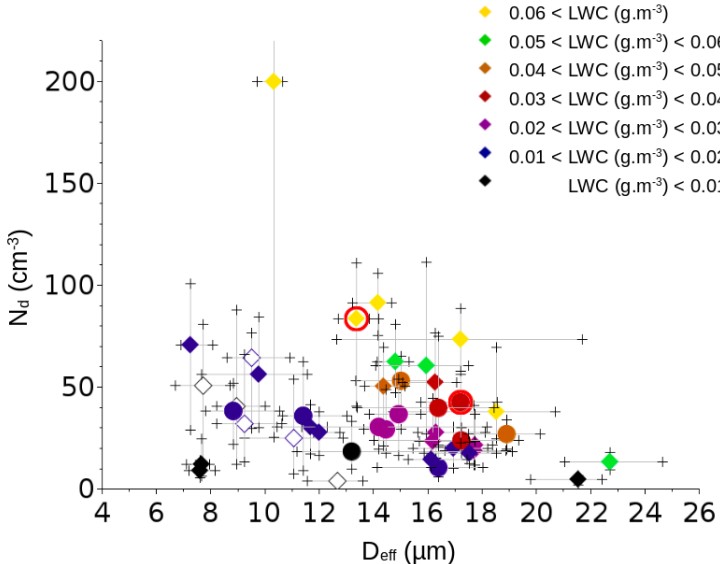

**Figure 1.** $N_d$ as a function of $D_{eff}$ for the 42 fog events. The colour set indicates the range of LWC values. Error bars are the 25th and 75th percentils. Statistics are made only when LWC > 0.005 g.m$^{-3}$. The diamonds and rounds represent fogs with one and two droplet mode, respectively. Thin fog are represented by empty diamonds. F9 is the yellow point surrounded by a red circle and F32 is the brown point surrounded by a red circle.

| Phase | $aN_{50^{th}}$ $(cm^{-3}.h^{-1})$ | $aL_{50^{th}}$ $(g.m^{-3}.h^{-1})$ | $aD_{50^{th}}$ $(\mu m.h^{-1})$ | $aT_{50^{th}}$ $(°C.h^{-1})$ |
|---|---|---|---|---|
| Formation | 23 | 0.029 | 12.2 | -0.39 |
| Development | 8.2 | 0.010 | 0.5 | -0.06 |
| Mature | -8.3 | -0.018 | -0.09 | 0.03 |
| Dissipation | -21 | -0.015 | 0.32 | 0.15 |

**Table 1.** Linear regression slopes of the temporal evolution of LWC, $N_d$, $D_{eff}$ and the temperature for the 4 phases identified in the 42 fog events.

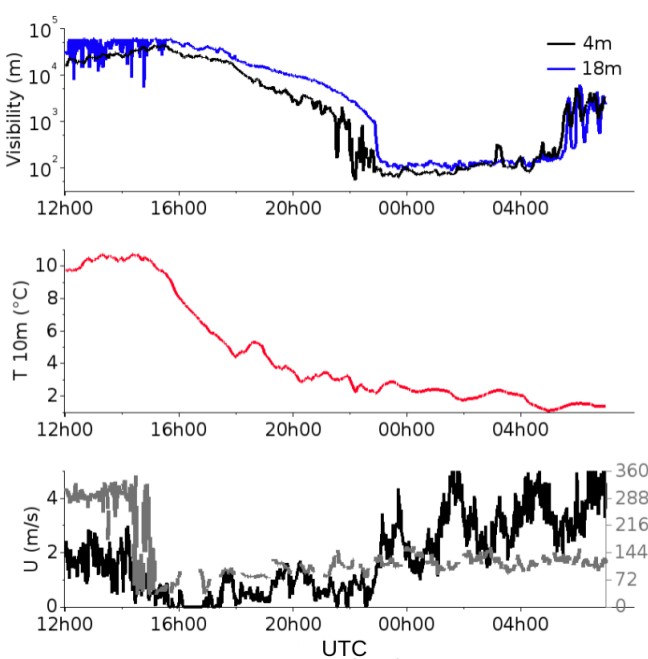

**Figure 2.** Temporal evolution of the visibility measured at 4 m and 18 m, the relative humidity, the temperature, the wind speed and the wind direction for F2.





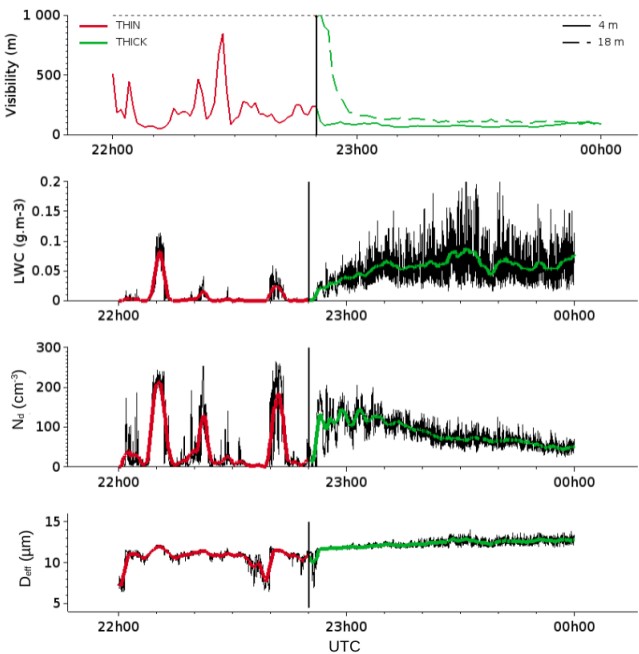

**Figure 3.** Temporal evolution of the visility, LWC, $N_d$ and $D_{eff}$ for F2 on [2-50] $\mu$m. Dark lines FM-100 data at 1s and colored lines at 1 min. Red points correspond to the thin phase and green point to the thick phase.

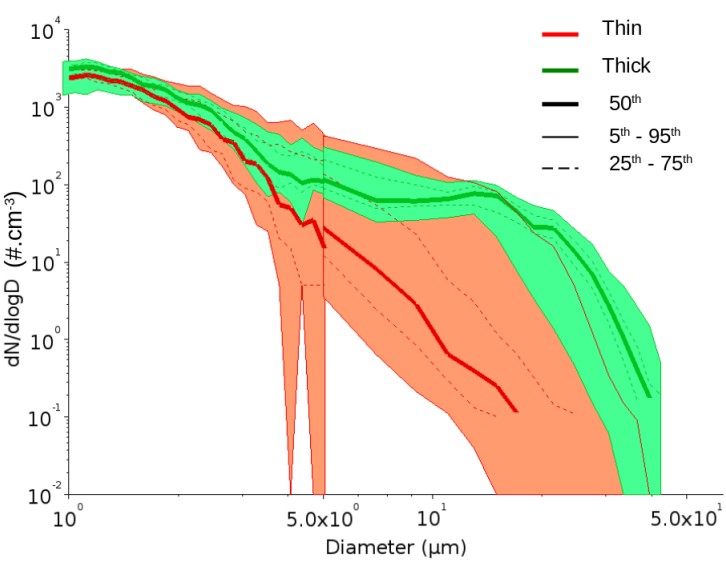

**Figure 4.** Median (bold lines) and filled values between percentils $5^{th}$ and $95^{th}$ for the aerosol and droplet size distributions during the vertical development of F2 for the periods of thin fog (in red) and of thick fog (in green).





**Figure 5.** Temporal evolution of the visibility measured at 4 m and 18 m, the temperature, the wind speed and the wind direction for (a) F9 and (b) F32.





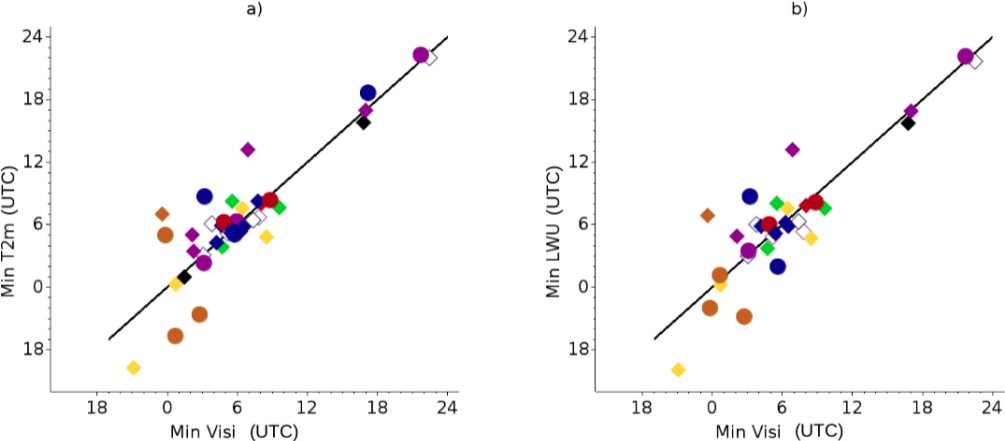

**Figure 6.** Correlation between the time of minimum visibility and (a) the time of minimum temperature at 2 m and (b) the time of upwelling infrared flux at 10 m for the 42 fogs events.



**Figure 7.** Temporal evolution of the visility, LWC, $N_d$ and droplet effective diameter (full lines) and droplet diameter percentil $5^{th}$ ,$25^{th}$ , $75^{th}$ and $95^{th}$ (dashed lines) for (a) F9 and (b) F32. The color coding is based on the separation of the fog event into four phase based on the evolution of the visibility. Dark lines represents the linear regression corresponding to each phase.



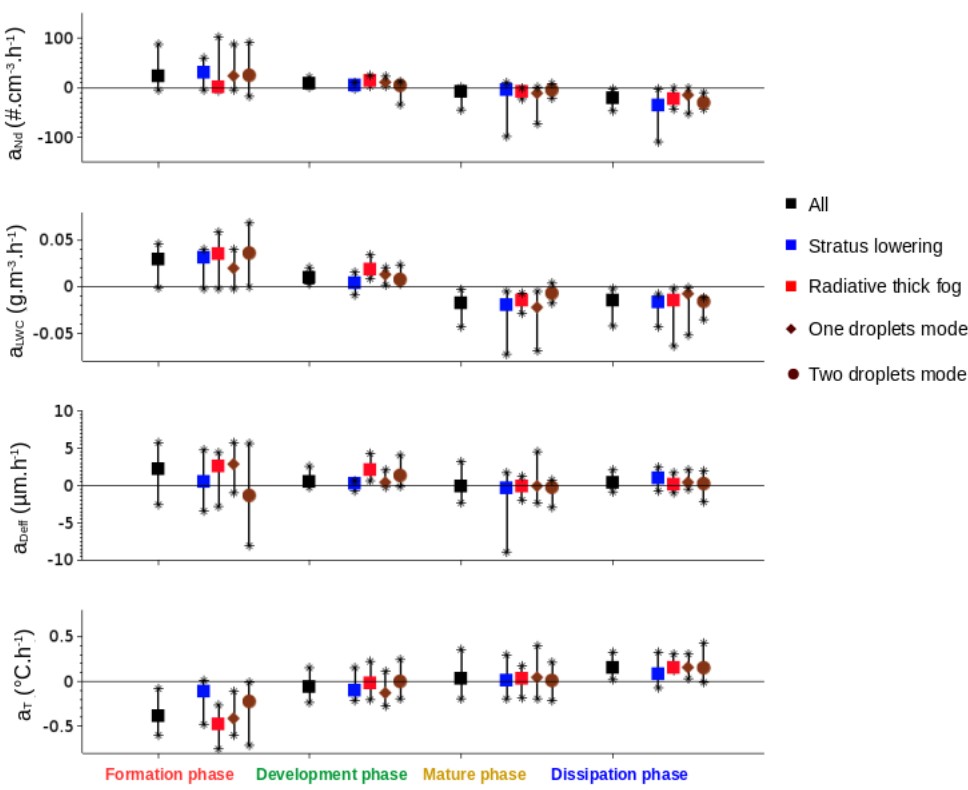

**Figure 8.** Statistical analysis of LWC, $N_d$ and $D_{eff}$ and the temperature at the different phases of the fog events for the fogs characterized by a single (grey squares), two droplet modes (grey circles) but also stratus lowering fog (blue marker) and radiative thick fog (red marker). Symbols represent the median values and error bars represent the 25th and 75th percentiles of the data.





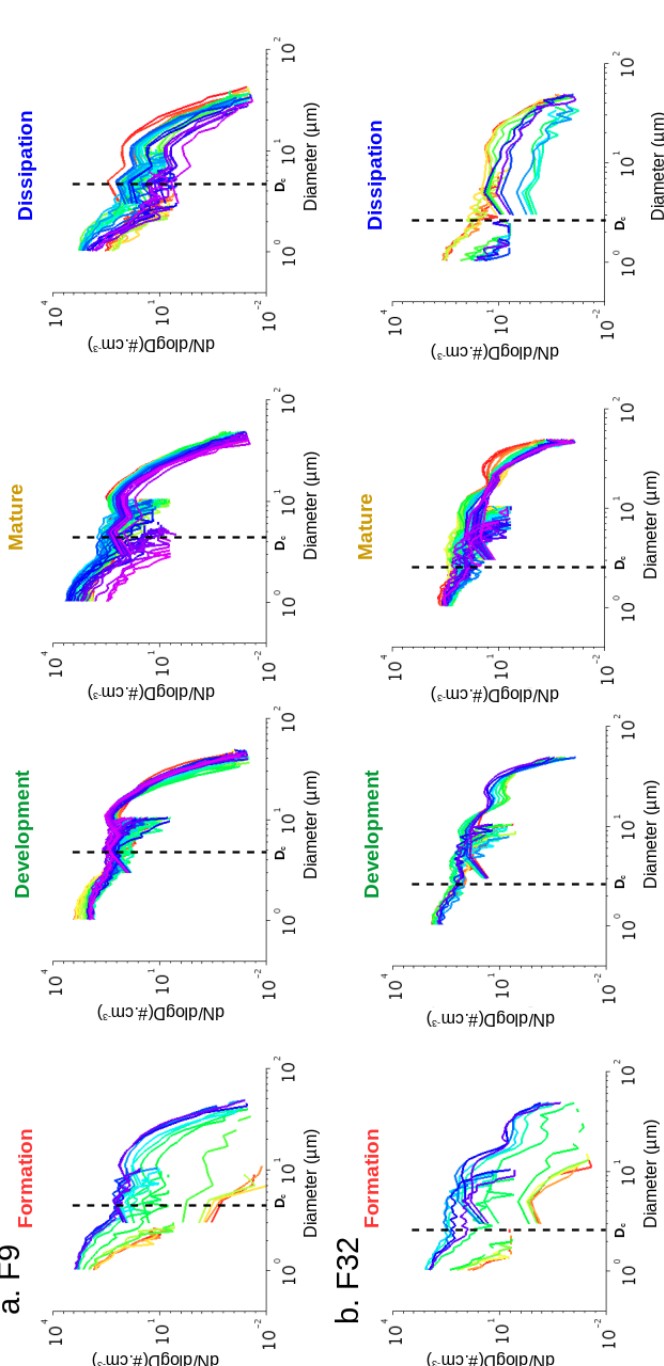

**Figure 9.** 5-min-averages of the aerosol and droplet size distributions during the fog life cycles of (a) F9 and (b) F32. Colors vary from red to violet according to time. Vertical dashed lines represents the activation diameter for each event as determined in Mazoyer et al. (2019).





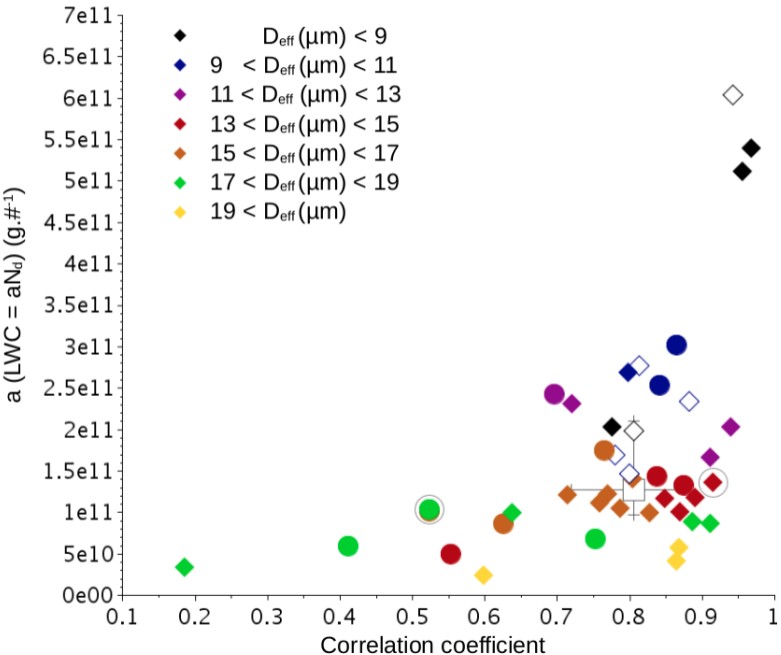

**Figure 10.** Correlation coefficient between $N_d$ and LWC for the 42 fog events. The colour set indicates the range of $D_{eff}$ values. Fogs with one and two droplet modes are denoted by the circle and diamonds markers, respectively. The square marker indicate the median value. F9 is the red point surrounded by a grey circle and F32 is the green point surrounded by a grey circle.

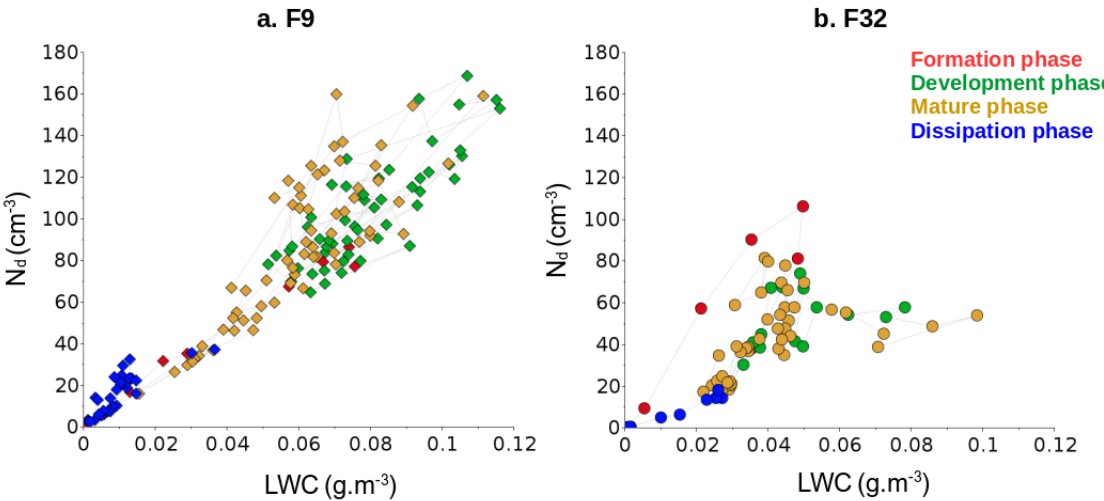

**Figure 11.** 5-min-averages of $N_d$ as a function of $D_{eff}$ for (a) F9 and (b) F32.





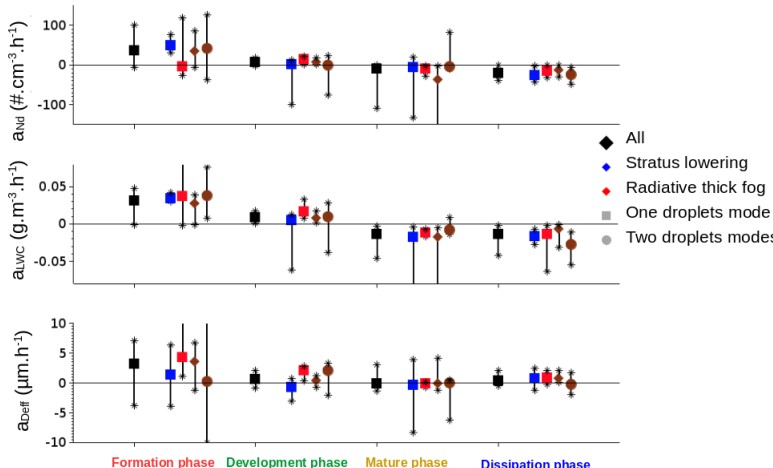

**Figure A1.** Same as Fig. 8 using wet critical diameter of the 23 fog of Mazoyer et al. (2019).

## Appendix A:  Temporal evolution of the 23 fog cases with determined wet critical diameter

## Appendix B:  Table of statistics on droplets distribution





| N° | | Initial time UTC | Final time UTC | Type 1 | Type 2 | Number of droplet modes | $N_d$ #.cm⁻³ 25th | $N_d$ 50th | $N_d$ 75th | LWC mg.m⁻³ 25th | LWC 50th | LWC 75th | $D_{eff}$ μm 25th | $D_{eff}$ 50th | $D_{eff}$ 75th |
|---|---|---|---|---|---|---|---|---|---|---|---|---|---|---|---|
| F1 | | 26/10/10 2:40 | 26/10/10 7:20 | RAD | THIN | 1 | 8 | 25 | 54 | 4 | 10 | 17 | 9 | 11 | 13 |
| F2 | f1 | 16/11/10 22:00 | 17/11/10 5:40 | RAD | THICK | 2 | 36 | 53 | 65 | 25 | 46 | 61 | 13 | 15 | 16 |
| F3 | f2 | 19/11/10 5:40 | 19/11/10 10:010 | STL | THICK | 1 | 23 | 52 | 77 | 25 | 39 | 52 | 15 | 16 | 17 |
| F4 | f3 | 19/11/10 15:40 | 19/11/10 17:50 | STL | THICK | 1 | 2 | 5 | 7 | 5 | 8 | 11 | 20 | 22 | 22 |
| F5 | | 01/11/11 22:45 | 02/11/11 8:00 | RAD | THICK | 1 | 56 | 74 | 89 | 58 | 96 | 195 | 13 | 17 | 22 |
| F6 | f4 | 10/11/11 18:00 | 11/11/11 17:30 | RAD | THICK | 1 | 21 | 50 | 70 | 17 | 44 | 57 | 14 | 14 | 15 |
| F7 | | 12/11/11 3:00 | 12/11/11 6:20 | STL | THICK | 1 | 47 | 63 | 81 | 39 | 58 | 80 | 14 | 15 | 15 |
| F8 | f5 | 15/11/11 2:30 | 15/11/11 9:40 | RAD | THICK | 1 | 17 | 30 | 41 | 9 | 15 | 20 | 11 | 12 | 12 |
| F9 | f6 | 16/11/11 1:10 | 16/11/11 13:30 | RAD | THICK | 1 | 41 | 84 | 111 | 37 | 64 | 77 | 13 | 13 | 14 |
| F10 | f7 | 16/11/11 16:00 | 17/11/11 0:10 | STL | THICK | 1 | 75 | 91 | 106 | 53 | 72 | 92 | 13 | 14 | 15 |
| F11 | f8 | 18/11/11 1:30 | 18/11/11 4:10 | RAD | THIN | 1 | 30 | 65 | 76 | 5 | 14 | 35 | 9 | 10 | 11 |
| F12 | f9 | 19/11/11 22:00 | 20/11/11 8:30 | RAD | THIN | 1 | 13 | 32 | 66 | 3 | 11 | 32 | 8 | 9 | 11 |
| F13 | f10 | 21/11/11 23:50 | 22/11/11 8:010 | RAD | THIN | 1 | 12 | 41 | 88 | 3 | 9 | 35 | 8 | 9 | 10 |
| F14 | f11 | 22/11/11 20:50 | 22/11/11 22:30 | RAD | THIN | 1 | 12 | 51 | 81 | 3 | 7 | 14 | 7 | 8 | 9 |
| F15 | f12 | 23/11/11 3:25 | 23/11/11 10:05 | RAD | THICK | 1 | 25 | 38 | 70 | 23 | 80 | 127 | 15 | 19 | 21 |
| F16 | f13 | 24/11/11 6:20 | 24/11/11 14:00 | STL | THICK | 1 | 12 | 19 | 41 | 16 | 26 | 48 | 17 | 18 | 19 |
| F17 | f14 | 24/11/11 16:10 | 24/11/11 18:15 | STL | THICK | 1 | 12 | 21 | 37 | 18 | 29 | 39 | 16 | 18 | 18 |
| F18 | f15 | 25/11/11 21:40 | 26/11/11 10:30 | STL | THICK | 1 | 27 | 61 | 111 | 28 | 50 | 97 | 14 | 16 | 18 |
| F19 | f16 | 28/11/11 6:30 | 28/11/11 10:40 | RAD | THICK | 1 | 10 | 13 | 18 | 42 | 50 | 74 | 21 | 23 | 25 |
| F20 | | 03/03/12 6:10 | 03/03/12 9:25 | STL | THICK | 1 | 10 | 20 | 30 | 11 | 15 | 24 | 16 | 17 | 18 |
| F21 | | 04/03/12 0:20 | 04/03/12 3:30 | STL | THICK | 1 | 15 | 28 | 40 | 13 | 25 | 29 | 16 | 16 | 18 |
| F22 | | 12/03/12 23:40 | 13/03/12 4:20 | RAD | THICK | 1 | 6 | 9 | 14 | 1 | 2 | 2 | 7 | 8 | 8 |
| F23 | | 14/03/12 3:40 | 14/03/12 6:40 | STL | THICK | 1 | 7 | 12 | 24 | 2 | 3 | 4 | 7 | 8 | 8 |
| F24 | | 16/03/12 4:37 | 16/03/12 6:50 | RAD | THICK | 1 | 29 | 71 | 101 | 5 | 14 | 16 | 7 | 7 | 8 |
| F25 | | 21/10/12 6:20 | 21/10/12 8:010 | RAD | THIN | 1 | 3 | 4 | 5 | 2 | 2 | 3 | 12 | 13 | 14 |




| N° | | Initial time UTC | Final time UTC | Type 1 | Type 2 | Number of droplet modes | $N_d$ #.cm⁻³ 25th | $N_d$ 50th | $N_d$ 75th | LWC mg.m⁻³ 25th | LWC 50th | LWC 75th | $D_{eff}$ μm 25th | $D_{eff}$ 50th | $D_{eff}$ 75th |
|---|---|---|---|---|---|---|---|---|---|---|---|---|---|---|---|
| F 26 | | 04/11/12 1:00 | 04/11/12 5:50 | RAD | THICK | 2 | 13 | 23 | 45 | 16 | 28 | 38 | 16 | 16 | 18 |
| F 27 | | 09/11/12 6:35 | 09/11/12 11:35 | RAD | THICK | 2 | 18 | 24 | 30 | 24 | 37 | 47 | 14 | 17 | 19 |
| F 28 | | 12/11/12 0:05 | 12/11/12 10:010 | STL | THICK | 2 | 16 | 37 | 54 | 14 | 24 | 39 | 14 | 15 | 16 |
| F 29 | f 17 | 16/11/12 20:45 | 17/11/12 9:20 | STL | THICK | 2 | 20 | 27 | 34 | 32 | 41 | 48 | 18 | 19 | 20 |
| F 30 | f 18 | 20/11/12 3:00 | 20/11/12 9:010 | RAD | THICK | 2 | 6 | 11 | 21 | 7 | 11 | 18 | 14 | 16 | 18 |
| F 31 | f 19 | 20/11/12 20:15 | 20/11/12 22:50 | STL | THICK | 2 | 17 | 29 | 49 | 16 | 22 | 38 | 14 | 15 | 15 |
| F 32 | f 20 | 22/11/12 3:15 | 22/11/12 9:010 | RAD | THICK | 2 | 23 | 43 | 58 | 29 | 38 | 47 | 16 | 17 | 19 |
| F 33 | f 21 | 30/11/12 19:00 | 01/12/12 2:45 | RAD | THICK | 2 | 21 | 40 | 75 | 20 | 38 | 76 | 13 | 16 | 18 |
| F 34 | | 01/12/12 4:40 | 01/12/12 10:25 | RAD | THICK | 2 | 10 | 18 | 33 | 4 | 10 | 31 | 11 | 13 | 19 |
| F 35 | | 01/12/12 16:25 | 01/12/12 18:45 | RAD | THICK | 2 | 9 | 36 | 63 | 5 | 14 | 19 | 11 | 11 | 13 |
| F 36 | f 22 | 10/01/13 2:15 | 10/01/13 3:36 | STL | THICK | 2 | 20 | 30 | 84 | 11 | 20 | 24 | 10 | 14 | 18 |
| F 37 | f 23 | 10/01/13 5:00 | 10/01/13 6:40 | STL | THICK | 1 | 13 | 18 | 23 | 14 | 17 | 21 | 16 | 18 | 19 |
| F 38 | | 12/01/13 5:15 | 12/01/13 6:40 | RAD | THICK | 1 | 16 | 28 | 38 | 8 | 17 | 23 | 12 | 12 | 13 |
| F 39 | | 21/01/13 23:25 | 21/01/13 23:55 | STL | THICK | 1 | 26 | 200 | 250 | 10 | 78 | 93 | 10 | 10 | 11 |
| F 40 | | 22/01/13 4:00 | 22/01/13 4:55 | STL | THICK | 1 | 11 | 14 | 17 | 10 | 13 | 17 | 15 | 16 | 17 |
| F 41 | | 23/01/13 0:25 | 23/01/13 8:25 | STL | THICK | 2 | 22 | 38 | 52 | 3 | 10 | 18 | 8 | 9 | 11 |
| F 42 | | 10/03/13 4:00 | 10/03/13 8:15 | RAD | THICK | 1 | 30 | 56 | 85 | 6 | 15 | 34 | 8 | 10 | 12 |

Table A1: Summary of information on the fog events analyzed in this study, including the fog event number (second column corresponds to fog studied in Mazoyer et al. (2019)) and corresponding date, the time span of the fog event, the type of the fog event (STL for stratus lowering fog, RAD for radiative fog, THICK for THICK fog and THIN for thin fog), the number of droplet mode and the 25th, 50th and 75th percentiles of $N_d$, LWC and $D_{eff}$ over fog life cycle as measured by the combination of the WELAS-2000 and the FM-100 on [3.79-50]μm range diameter.





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
