# Peer review of "Experimental study on the evolution of droplets size distribution during the fog life cycle"

_Atmospheric Chemistry and Physics, 2021_

## Author Comment (AC1)

Replies to the Editor and reviewers are in blue. Reviewers' comments are in black.

The authors would like to thank the reviewers for their helpful and interesting comments on the manuscript. Hereafter we answer all the comments. Our responses to the reviewers' comments appear in blue. Please also note that we have added more information on our fog selection according to the Tardif and Rasmussen classification of 2007, a discussion on non-local effects for fog development, as well as a discussion on the effects of turbulent motions on activation. Modifications in the main text appear in red.

**REVIEWER 1 :**

General

This paper examines a reasonably large data set of microphysical measurements made in fog at the SIRTA observatory on the outskirts of Paris, and describes the microphysical properties of those fogs, suggesting some implications for the numerical prediction of fogs.The data set is of sufficient size to add useful knowledge to the literature and help to characterize surface conditions in fogs, and also support numerical studies of fog. However, I would like to see more analysis on the dynamic effects of meteorology on the fog, in particular the effects of larger scale non-local dynamics. In Addition, certain important aspects of fog development do not appear to be discussed properly.

Authors thank the reviewer for its encouraging comments. Some modifications have been made to the text following the reviewer recommendation. In particular, clarifications have been made concerning the fog types we sampled in order to eliminate advection fog, we followed the Tardif and Rasmussen classification of 2007. Since a slight advection could have been present for some of our events we have also added a discussion on the non-local effect and moderate our text.

Main comments

Firstly, a fuller description of the SIRTA site and its surroundings would be useful (e.g. show a topographic map?). It is well known that stable boundary layer flows and fog are affected by topography, and at SIRTA we have the possible further effect of a change in roughness length between the urban and rural environments. These characteristics may affect the fog development and I believe should be discussed.

This leads onto my main point, which is that it has been well demonstrated in the literature that fog development is very often profoundly affected by larger scale (non-local) effects (indeed the authors refer to Duconge et al who discuss these in some detail), yet these do not appear to be considered much in this analysis. Instead, it appears to assume that all the fogs analysed have developed locally. However, examining data presented in their figures indicates there is likely to be a non-local element in their evolution.

To provide a better description of the SIRTA site we added in the part 'Observational site and instrumentation' the following sentences: 'The site is installed in a semi-urban area with mixed

land-cover including forest, lake, meadows and shrubs next to a built-up area. It is located on a plateau elevated 60 m higher than the surroundings.'

Duconge et al. (2020) showed that non local effects are as important as local effects in valley fog in their specific case, the wide and open valleys induce a specific circulation with slope wind. They showed that wider basins are more subject to local formation of dense fogs than narrow valleys where advecting fog events are more frequently observed. a. In addition, to avoid considering potential non-local effects (Duconge et al., 2020), only radiative and stratus lowering fog have been selected in our analysis. ,

A discussion on that topic has been included in section Discussion: 'A recent study by Duconge et al. (2020) has shown that non-local effects can be as important as local effects in valley fog development over the Shropshire hills(Duconge et al., 2020). Therefore, non-local effects due to either topography or surface heterogeneities or due to the mixed-land cover around the SIRTA plateau may have an impact on the SIRTA fog thickening and formation at the SIRTA station. However, to avoid considering non-local effects, only radiative and stratus lowering fog have been selected in our analysis.'

For example, both figure 2 and figure 5 show sharp discontinuities in visibility, temperature (some converging to adiabatic from stable) and often winds, at fog onset. This is normally a characteristic of a well developed fog that has advected over the observation site, usually in the form of a gravity current. Clearly these events can complicate the interpretation of microphysical data and therefore should be discussed. If the authors wish to discuss the in situ evolution of microphysical processes, then clearly, choosing cases from their database least affected by advection would be sensible.

In relation to the above, I believe that a common mis-interpretation of the data is that it is sometimes reported that the appearance of stronger winds and turbulence is responsible for the vertical development of shallow fog. However, in these cases, examination of larger scale data, when available, normally shows that the vertical development is associated with advection of a deeper fog over the site, bringing stronger winds and turbulence with it (i.e. the casual effect is opposite to that proposed). I believe this may be the case for F2, presented on page 6 (line 10).

In figure 2 and 5 we observed two significant differences when comparing thin and thick fogs: 1/ Microphysical properties are evolving between thin-to-thick transitions and 2/ Transitions are associated with a change in dynamical structure. In cases where, despite our fog selection to avoid the possible influence of non-local effects, fog advection would have enhanced the thickening of the fog layers, this can't explain the growth processes we specifically observed during the thin-to-thick transition. However, it appears to us that an advection of a deeper fog due to a wind increase is not in opposition with the fact that fog development and wind increase are linked. Our affirmation is that without a little push up of turbulence, shallow fog stays shallow.

To mitigate our text, we added in the 'Thin-to-thick transition' subsection :'A gravity current could be responsible for the association of fog development and wind increase (Bardoel et al., 2021).

However, the data selection we performed in section Fog type and classification should have limited the influence of non-local effects on fog development.'.

Following on from these points I am not sure that adhering to the four-stage fog model presented (formation, development, mature, dissipation) is very helpful. For advecting fog cases it is misleading to label the arrival period as 'development' since that occurred elsewhere previously. Similarly for lowering stratus. I believe it would be more useful to relate microphysical changes to dynamic or thermal properties or their changes, within the fog. An example would be for the dissipation phase. Fog may dissipate through a variety of mechanisms; mixing, radiation, sensible heating, so it seems sensible to try to relate microphysical changes to these processes. It can also be noted that we expect to see microphysical changes at the surface due directly to dynamical processes. For example, in cases where a shallow stable fog evolves into a deeper adiabatic one, we normally see a significant change in the droplet spectra measured at the surface. This is due to the fact that in a stable fog, the greatest LWC is normally near the surface, but once transformed into an adiabatic fog the maximum LWC is higher up and the region next to the ground is normally then the least saturated region of the fog. Thus it is common to see a decrease in LWC (and sometimes $N_d$ ) at this transition. For these reasons I believe the paper requires a much more in-depth discussion of fog dynamics in order to interpret the microphysical data.

Once again this study had only selected radiative fog and stratus lowerings ones according to the Tardif and Rasmussen classification (2007). We are very sorry that this did not appear clearly in the first version of the paper and we are grateful to the reviewer for highlighting it.
Then concerning the dissipation phase, we did not try to explain the dissipation mechanisms in the text but how the microphysics is evolving during this stage. We agree with the reviewer that LWC is increasing at upper level because of the adiabatic evolution of the fog. However LWC decreases at lower levels are linked to other processes. Surface warming because fog emissivity can be stronger than surface one is an example. This kind of radiative effects are supported by Figure 6.

As a consequence, my feeling is that some of the conclusions made are not really supported by the analysis.
We hope that the modification we made to the paper better supports our conclusions.

Regarding the observations there is no discussion of instrument errors or characteristics. Considering the Welas-2000, can the authors guarantee that the hydration of aerosols has not changed between ambient and sampling environments? I presume that any significant changes in hydration will have undesirable consequences for your analysis?
The WELAS sensor is commonly used for measuring non-hydrated aerosol particles, hydrated aerosol particles and activated fog droplets (Haeffelin et al. 2010 ; Denjean et al 2014; Rosati et al. 2014 ; Elias et al. 2015 ; Mazoyer et al. 2019). Denjean et al (2014) showed the ability of the WELAS sensor to measure the hygroscopic growth of laboratory-generated aerosol particles up to 95% RH without any drying artifact. During the ParisFog field campaign, the instrument was

installed at the top of the container without passing through the aerosol inlet to prevent the drying of the sampling air masses.

Also, the 'wet-critical' threshold of 3.79 microns seems rather precise, given the accuracy of sizing of the spectrometers (for the F100 I expect it will be a micron or two)?
Well, such a precision comes from the Mazoyer et al, 2019 studies on 42 fog events. It corresponds to a median value. Our willingness is to use a unique value for our whole set of events. We change 3.79 microns for 3.8 microns.

**REVIEWER 2 :**

General
The study presents a set of observations on fog microphysics, and analyses how droplet microphysics depends on the fog formation method and also the fog lifecycle. This is one the largest datasets presented for the fog properties, and thus provides a nice addition into previous data presented in literature. The study also fits well in the scope of ACP. I agree with the Referee #1 related to analysis between local and non-local effects, and more detailed analysis would make the study stronger. Below are some additional comments and suggestions to help clarify some points. After those have been considered, the manuscript can be accepted for publication in ACP.
Authors thank the reviewer for its encouraging comments. Sentences have been added on the possible impact of turbulent motion on new aerosol activation. We notice the strong interest of the reviewer for more vertical analysis of the fog, unfortunately the ParisFog set-up was calibrated for a quasi 0D analysis, hopefully forthcoming studies (on SOFOG for example) should bring new insights on vertical processes.

Main comments
I would like to see more detailed analysis especially related to the presence of cloud droplets with two modes. Can it be related to stability within fog or aerosol concentration? For example, do you trust enough to the temperature observations and could the temperature difference between 1m and 30m be used to estimate stability?

Abstract, line 10: This statement should be studied in more detail, please see the comments below.
We agree with the reviewer that more detailed analysis related to the presence of cloud droplets with two modes would be tremendously interesting. Unfortunately, investigating the link between fog and aerosol concentration is not possible because the SMPS did not capture aerosols larger than 0.5 µm, which could contribute to the fog droplets in the second mode. During the fog development and mature phase, temperature at 1m and 30m are often equal as shown in figure5.

Abstract, line 14: I'm not sure if I understand this correctly but isn't the surface also warmed because of decreased radiative cooling from the surface as fog grows optically thick, and then the heat from the ground is starting to produce sensible heat flux.

Yes, this is right. Fog emissivity being higher than surface ones, surface stops to cool and produces sensible heat flux.

Abstract, lines 17-18: "Although a positive relationship is found in most of the events due to continuous activation of aerosol into fog droplets" Please see later comments related to this statement.

We agree with the reviewer. However, models do not perform continuous activation even when used with sophisticated schemes for supersaturation calculation (Mazoyer PhD, not published in english). Consequently it seems very important to us to highlight it.

We made the following change in the 'Temporal evolution of microphysical properties' section for:

'Droplet number concentration is still increasing in the development phase as more is formed than lost in fog processing and sedimentation, causing a slight increase of Nd and Deff although their production and growth are much slower. '

Abstract, lines 19-21: I agree with this!

We gratefully thank the reviewer for encouraging us in this direction.

Page 2, line 2: Reference to Boutle et al (2021) can be now updated to final published version.

Corrected.

Pager 4, lines 24-27: The definition of optically thin and thick fog seems to be related to fog actual height here and thus local property, and not to the fog actual optical depth or change in atmospheric stability and turbulent mixing within the fog. Please formulate this more clearly.

We have made the following changes :'Following Elias et al. (2009)  and Dupont et al. (2016), further classification of radiation fogs was performed based  on their vertical  development using the comparison of the two diffusometers: a thick fog produces low-visibility conditions at simultaneously 4 and 18 m, whereas a thin fog  produces low-visibility conditions at 4m only.'

Page 5, line 21: "*These results confirm that the predictability of droplet activation in fog can not be described by LWC only.*" I don't understand this statement as droplet number concentration is more likely to drive LWC in fogs than opposite.

Mazoyer et al. (2019) showed that no direct correlation could be observed between Nd and LWC. In the present paper Figure 1 also shows no direct correlation between these two quantities. We believe that activation (Nd) is linked to peak in supersaturation and that LWC is
linked to supersaturation in its temporal integral evolution. Thus they do not depend on the exact same process.

Page 5, line 27: 46 g/m3 is probably wrong for liquid water content.
Corrected to 46 mg/m3. Thanks.

Page 6, line 5: Radiation fog can develop in height also under low wind speed, but wind and turbulence caused by surface can certainly fasten the growth. But, if the wind speed is too strong, radiation fog can't form at all.
Exactly, only dew would be observed in such a case.

Page 7, line 17: At what time did the fog form at 150m and how quickly did it lower to be a fog?
One of the features of this event was the initial formation of a cloud layer at 150 m a.g.l., followed 30 min later by fog occurring at the surface. As underlined by Stolaki et al. (2015), this characteristic is very common at SIRTA and 88 % of the radiation fog events dur-ing the field experiment followed a similar pattern. However,these events are not classified as stratus lowering as they were followed rapidly by formation of fog at the surface. A delay of 30 min between the formation at 150 m height and at the ground seems too short to be a stratus lowering, which is mainly driven by the evaporation of slowly falling droplets that cool the sub-cloud layer (Dupont et al., 2012). This suggests that this type of radiation fog could be linked with, and specific to, the configuration of the SIRTA site.
At 150 m the F9 fog forms approximately 30 min before reaching the ground according to the radar measurements.

Page 8, line 24: New droplets are formed through most of the fog lifecycle, but maybe you should state that the droplet number concentration is just increasing as more is formed than lost in fog processing and sedimentation.
We agree with the reviewer. We change the sentence in the text to: 'Droplet number concentration is still increasing in the development phase as more is formed than lost in fog processing and sedimentation, causing a slight increase of Nd and Deff although their production and growth are much slower. '

Page 9, lines 19-21: This is true, but behind the process should be a very stable fog without turbulence. In such conditions the new droplet formation is taking place at the top of fog in the layer radiatively cooling, and thus the fog droplet size distribution is developing as described and observed in the lower parts of the fog.
According to Xue et al. (2008), collision-coalescence in fog is enhanced by turbulent motion. For what concern Ostwald ripening, Yang et al. (018) also put forward a higher efficacy associated with turbulent motion. Moreover it seems to us that very stable fog without turbulence does not exist. Once the radiative cooling had taken place at the fog top, top-bottom subsidence movement took place, due to a kind of convective movement but enhanced by the colder top and turbulent motions follow.

Page 10, lines 16-21: "*The increase of LWC with increasing Nd in most of the fog events is mainly due to the continuous activation of aerosol into droplets (i.e. section 3.4)*". I would state this in other way. The droplet size in stable fog is limited by sedimentation of larger droplets. Thus, after growing to large enough sizes droplets are simply removed from the system and thus the liquid water content is quite strongly correlated with droplet number concentration especially close to the surface. Situation could be different if liquid water content and number concentration are compared at the altitude where droplets are forming inside the fog.

Our concern is to highlight the continuous activation during the fog life cycle because this process is not currently represented in the model. We agree with the reviewer that both processes - loss of LWC by sedimentation and increase of Nd by activation - are important. However, Bott et al., 1990 showed that the loss of LWC by sedimentation constitutes a loss of the sink terms for vapor deposition and an opportunity to temporarily increase supersaturation.

We are also quite excited to see forthcoming results at higher altitude.

Page 10, lines 16-21: "*It is plausible that the low supersaturation limits the growth of droplets by condensation and the consumption of the water content. The excess water vapor could therefore become available for additional activation of aerosols into cloud droplets.*" If the supersaturation is low enough to prevent growth of droplets, it is certainly also low enough to prevent the activation of new droplets unless there is some source of efficient CCN. It is plausible that droplet number is so low, that if there is turbulence within the fog, some aerosols can be activated. But turbulence can form only after fog has grown into unstable phase as described in Boutle et al (2018), or due to some local topography effect.

We agree with the reviewer that excess supersaturation can be brought by turbulent motions or local topographic effects as well.

We have added this information as follows: 'However, once the fog has grown into the development phase, turbulent motions could contribute to bring supersaturation to peak values (Boutle et al., 2018) as well and to new aerosols activation .'

Page 10, Discussion: The selection of wet critical diameter is difficult, and it is also complicated by the fact that hydrated aerosol particles can also reduce the visibility within the fog. So in some conditions the knowledge of aerosol is also needed to be able to forecast visibility, as using the actual fog droplet number concentration can lead to too high visibility forecast.

We agree that an accurate prediction of the visibility should also consider the hydrated aerosol evolution. However, for what concerns the physical process affecting droplet evolution it seems fundamental to separate hydrated aerosols from droplets. It is why we use a wet critical diameter.

Figure2: Related to connection of wind speed and droplet growth, it would be interesting to see wind speed and direction also at the higher observation altitude. It could give hint if it is turbulence that is enhancing fog growth, or opposite and growth will increase turbulence. Also, as you have data up to 30m, please show it also here. In case it is local radiation fog it should grow steadily in

height, but in case the transition at different altitudes is taking place simultaneously, there is more likely some advection affecting the observations.

Unfortunately, wind measurements were not available at 30m during this field campaign.

In response to the other reviewer we have mitigated our text and added some sentences on the non-local processes that can affect fog development. However the selection we made on the studied fog was done according to the Tardif and Rasmussen algorithm and should concern only radiation and stratus lowering fog.

Bardoel, S. L., Horna Muñoz, D. V., Grachev, A. A., Krishnamurthy, R., Chamorro, L. P., & Fernando, H. J. (2021). Fog formation related to gravity currents interacting with coastal topography. Boundary-Layer Meteorology, 181(2), 499-521.

Bott, A., U. Sievers, et W. Zdunkowski, 1990 : A radiation fog model with a detailed treatment of the interaction between radiative transfer and fog microphysics. Journal of the atmospheric sciences, 47 (18), 2153–2166.

Boutle, I., Price, J., Kudzotsa, I., Kokkola, H., and Romakkaniemi, S.: Aerosol–fog interaction and the transition to well-mixed radiation fog, Atmospheric Chemistry and Physics, 18, 7827–7840, 2018

Ducongé, L., Lac, C., Vié, B., Bergot, T., & Price, J. D. (2020). Fog in heterogeneous environments: the relative importance of local and non-local processes on radiative-advective fog formation. Quarterly Journal of the Royal Meteorological Society, 146(731), 2522-2546.

Dupont, J., Haeffelin, M., Stolaki, S., and Elias, T.: Analysis of dynamical and thermal processes driving fog and quasi-fog life cycles using the 2010–2013 ParisFog dataset, Pure and Applied Geophysics, 173, 1337–1358, 2016.

Elias, T., Haeffelin, M., Drobinski, P., Gomes, L., Rangognio, J., Bergot, T., Chazette, P., Raut, J.-C., and Colomb, M.: Particulate contribution to extinction of visible radiation: Pollution, haze, and fog, Atmospheric Research, 92, 443–454, 2009.

Elias, T., Jolivet, D., Dupont, J.-C., Haeffelin, M., and Burnet, F.: Preliminary results of the PreViBOSS project: description of the fog lifecycle by ground-based and satellite observation, in: SPIE Remote Sensing, pp. 853 406–853 406, International Society for Optics andPhotonics, 2012.

Mazoyer, M., Lac, C., Thouron, O., Bergot, T., Masson, V., & Musson-Genon, L. (2017). Large eddy simulation of radiation fog: impact of dynamics on the fog life cycle. Atmospheric Chemistry and Physics, 17(21), 13017-13035.

Mazoyer, M., Burnet, F., Denjean, C., Roberts, G. C., Haeffelin, M., Dupont, J.-C., and Elias, T.: Experimental study of the aerosol impact on fog microphysics, Atmospheric Chemistry and Physics, 19, 4323–4344, doi:10.5194/acp-19-4323-2019, https://www.atmos-chem-phys.net/19/4323/2019/, 2019.

Tardif, R. and Rasmussen, R.: Event-based climatology and typology of fog in the New York City region, Journal of applied meteorology and climatology, 46, 1141–1168, 2007

*Xue, Y., L.-P. Wang, et W. W. Grabowski, 2008 : Growth of cloud droplets by turbulent collision-coalescence. Journal of the Atmospheric Sciences, 65 (2), 331–356.*

*Yang, F., Kollias, P., Shaw, R. A., and Vogelmann, A. M.: Cloud droplet size distribution broadening during diffusional growth: ripening amplified by deactivation and reactivation, Atmospheric Chemistry and Physics, 18, 7313–7328, 2018.*

---

## Author Response (AR2)

Replies to the Editor and reviewers are in blue. Reviewers' comments are in black.

The authors would like to thank the reviewers for their helpful and interesting comments on the manuscript. Hereafter we answer all the comments. Our responses to the reviewers' comments appear in blue. Please also note that we have mitigated our discussion on non-local effects for fog development. Also we have added more information on the cause of formation of radiation fog at elevated level and a new paragraph in section 4 on the need to improve our knowledge on the fog life cycle along the vertical.
Modifications in the main text appear in red.

**REVIEWER 1 :**

Authors state in the updated manuscript that "The site is installed in a semi-urban area with mixed land-cover including forest, lake, meadows and shrubs next to a built-up area. It is located on a plateau elevated 60 m higher than the surroundings." and in the reply for the Reviews "Moreover it seems to us that very stable fog without turbulence does not exist." Does this actually mean that because of the local topography, you cannot observe the development of radiation fogs from the first steps at SIRTA station, as fog is first formed in the surroundings, and it is already developed when the top reaches the plateau 60m higher than surrounding? This naturally depends on the size of plateau and wind speed/direction, but could certainly explain the lack of stable fogs. Because of this addition, I feel a more detailed information is needed connecting plateau size, wind speed and turbulent intensity affecting the droplet activation if such is not presented in earlier studies related to SIRTA observations.

Radiation fog events initially formed at the surface are observed at SIRTA (Stolaki et al., 2015 ; Mazoyer et al., 2017). As already mentioned in section 3.3 of the present paper, this characteristic represents 12% of the radiation fog events sampled during the field campaign. The other 88% were initially formed at elevated level (~150 meter above ground level), and reached the ground in a very short time (<30 minutes). Mazoyer et al. (2017) demonstrated that it is a consequence of the tree drag effect (and not local topography) when the wind meets this obstacle and the deposition effect, which reduces the formation of droplets near the surface. The differences in terms of meteorological and microphysical properties of two contrasted radiation fog events formed at the surface (F32) and in altitude (F9) are presented in detail in sections 3.3, 3.4 and 3.5.

We agree with the reviewer that further study should investigate deeply the relation between droplet activation, droplets evolution and turbulent intensity. Turbulence in radiation fog is mainly driven by fog-top cooling and by vertical wind shear (e.g. Figure 2 of Yang et al., 2020 for instance). This means that vertical wind and turbulent motion profiles are needed to investigate the origin of turbulence in fog and its impact on fog evolution. However, the experimental set up of the ParisFog field campaigns do not allow us to investigate these processes.

Following the the reviewer's comments, we have added :

- more information in section 3.3 on the cause of formation of radiation fog at elevated level : « Mazoyer et al. (2017) demonstrated that it is a consequence of the tree drag effect (and not local topography) when the wind meets this obstacle and the

deposition effect, which reduces the formation of droplets near the surface. »

- a new paragraph in section 4 on the need to improve our knowledge on the fog life cycle along the vertical : « The study presented in this paper focused on the fog life cycle based on ground-based observations. Bergot et al. (2015) and Mazoyer et al. (2017) showed that surface heterogeneities can induce significant variabilities in the vertical distribution of the fog layer. Waersted et al. (2017) observed using remote sensing instruments a critical role of vertical structures in the fog layer in controlling fog top radiative cooling. Recent studies have underlined the necessity to add a detailed representation of activation processes along the vertical (Egli et al., 2015; Stolaki et al., 2015, Mazoyer 2016). Further field investigations of the vertical distribution of fog meteorological and microphysical properties are required to provide insight on the microphysical processes driving fog variability and the relationship between turbulence,radiation, droplets activation and droplets evolution in order to improve the representation of processes parameterization of fog events by NWP models.

**REVIEWER 2 :**

I am still not entirely convinced that the arguments relating to advection are valid. For example, the authors appear to be claiming that radiation fogs do not advect but this is not the case! On balance however I think there is enough useful material (microphysics observations) to merit publication.

We agree with the reviewer that radiation fogs can be advected. However, as stated by Ducongé et al, 2020, local processes are balanced by non-local processes. In some configurations local effects are more important than non-local ones and a good understanding of microphysics becomes a requirement.

As our text was not clear, sentences in the discussion session had been modified to :
To limit the impact of non-local effects, only radiative and stratus lowering fog have been selected in our analysis.
Also sentence have been added in a new paragraph in section 4:
Further field investigations of the vertical distribution of fog meteorological and microphysical properties are required to provide insight on the microphysical processes driving fog variability and the relationship between turbulence,radiation, droplets activation and droplets evolution in order to improve the representation of processes parameterization of fog events by NWP models.
And in the Fog type and classification session to:
To minimize the impact of non-local effects (Ducongé et al, 2020), only radiative and stratus lowering fog have been selected.

**References**

*Ducongé, L., Lac, C., Vié, B., Bergot, T., & Price, J. D. (2020). Fog in heterogeneous environments: the relative importance of local and non‑local processes on radiative‑advective fog formation. Quarterly Journal of the Royal Meteorological Society, 146(731), 2522-2546.*

*Mazoyer, M., Lac, C., Thouron, O., Bergot, T., Masson, V., & Musson-Genon, L. (2017). Large eddy simulation of radiation fog: impact of dynamics on the fog life cycle. Atmospheric Chemistry and Physics, 17(21), 13017-13035.*

*Mazoyer, M., Burnet, F., Denjean, C., Roberts, G. C., Haeffelin, M., Dupont, J.-C., and Elias, T.: Experimental study of the aerosol impact on fog microphysics, Atmospheric Chemistry and Physics, 19, 4323–4344, doi:10.5194/acp-19-4323-2019.*

*Yang, Y., & Gao, S. (2020). The impact of turbulent diffusion driven by fog‑top cooling on sea fog development. Journal of Geophysical Research: Atmospheres, 125(4), e2019JD031562.*